# Multi-omics single-cell analysis reveals key regulators of HIV-1 persistence and aberrant host immune responses in early infection

Dayeon Lee[1], Sin Young Choi[1], So-I Shin[1], Hyunsu An[1], Byeong-Sun Choi[2]*, Jihwan Park[1]*

[1]School of Life Sciences, Gwangju Institute of Science and Technology, Gwangju, Republic of Korea; [2]Director for Laboratory Diagnosis and Analysis, Chungcheong Regional Center for Disease Control and Prevention, Korea Disease Control and Prevention Agency (KDCA), Daejeon, Republic of Korea

*For correspondence:
byeongsun@korea.kr (B-SC);
jihwan.park@gist.ac.kr (JP)

## eLife Assessment

This study presents **important** findings that enhance our understanding of immune cell interactions in the context of chronic HIV-1 infection. The evidence supporting the conclusions is **convincing**. The authors have employed appropriate and validated methodologies, including detailed data reprocessing and batch correction to account for inter-donor variability. The inclusion of supplementary figures and analyses, such as cell communication inference, further substantiates the robustness of the findings. Overall, this work contributes to our understanding of HIV-1 immune evasion and highlights potential therapeutic targets for reservoir eradication.

**Abstract** The clearance of human immunodeficiency virus-1 (HIV-1) remains a significant public health challenge due to impaired cellular immune responses and HIV-1 maintenance during acute infection. However, the genetic and epigenetic changes influencing the immune response on host infected cells remain unclear. Here, this study analyzes HIV-1-infected CD4+ T cells from peripheral blood mononuclear cells from people living with HIV-1 during early infection (<6 months) using single-cell RNA and ATAC sequencing. It is observed that HIV-1 hinders the antiviral response, particularly by interfering with the interferon signaling pathway. Multimodal analysis identifies KLF2 as a key transcription factor in infected CD4+ T cells. Moreover, cells harboring HIV-1 provirus are predominantly identified as Th17 cells, which exhibit elevated KLF2 activity. This suggests an increased susceptibility to HIV-1 infection and a constrained immune response due to the quiescent characteristics of these cells. The finding provides insights into the immune mechanisms and key regulators of HIV-1 maintenance in CD4+ T cells during the early stages of infection.

## Introduction

Each year, approximately 2 million people are infected with human immunodeficiency virus-1 (HIV-1), and about 1.1 million individuals die from HIV-related illnesses. HIV-1 primarily targets CD4+ T lymphocytes, leading to a gradual depletion of this critical T cell population and the eventual onset of acquired immunodeficiency syndrome (*Douek et al., 2003*; *Pantaleo et al., 1993*). Although antiretroviral therapy (ART) helps inhibit HIV-1 replication (*Permanyer et al., 2012*), the persistent presence of the virus causes chronic inflammation, contributing to HIV-related morbidity and mortality

(*Peterson and Baker, 2019*; *Zicari et al., 2019*). Additionally, latent proviruses can be reactivated and cause viral rebound if ART is discontinued. As a result, achieving complete clearance of HIV-1 remains a major goal and public health challenge (*Battistini and Sgarbanti, 2014*).

During HIV-1 infection, cellular immune responses are disrupted, leading to immune cell dysfunction and alterations in the immune cell landscape. Acute HIV infection (AHI) occurs from 3 weeks to 6 months after transmission, following an initial period when the viral load is undetectable (*Fiebig et al., 2003*). Once proviruses integrate into the host genome, they impact the recruitment of regulatory factors, thereby affecting gene expression (*Verdikt et al., 2021*; *Lange et al., 2020*; *Park et al., 2014*). These modifications during AHI play a role in the host cell's initial immune response and contribute to HIV-1 latency (*Battistini and Sgarbanti, 2014*; *Parker et al., 2023*; *Deeks et al., 2015*). Studying transcriptomic changes during AHI is crucial to identify primary immune responses and regulatory factors that influence HIV-1's effect on host immune cell function.

Single-cell RNA sequencing (scRNA-seq) and single-cell Assay for Transposase Accessible Chromatin with high-throughput sequencing (scATAC-seq) are powerful techniques that provide insights into cellular diversity, allowing for high-resolution analysis of gene expression and regulatory landscapes in individual cells (*Yoon et al., 2024*; *Bae et al., 2024*; *Eun et al., 2024*; *Kim and Park, 2021*; *Luo et al., 2022*; *Heumos et al., 2023*). Recent advancements in sequencing technologies have driven numerous studies to uncover biological factors contributing to HIV-1 pathogenesis (*Lee et al., 2021*; *Bradley et al., 2018*; *Bruner et al., 2019*; *Kazer et al., 2020*; *Sun et al., 2023*; *Wei et al., 2023*). However, challenges remain in studying genetic and epigenetic regulation in HIV-1-infected cells: low HIV-1+ CD4+ T cell frequency in peripheral blood mononuclear cells (PBMCs) limits reliable analysis (*Bruner et al., 2019*; *Ho et al., 2013*), ex vivo activation in previous studies does not capture in vivo heterogeneity (*Bradley et al., 2018*; *Liu et al., 2020*; *Golumbeanu et al., 2018*). Additionally, although recent studies have attempted direct ex vivo omics analyses (*Collora et al., 2022*; *Wu et al., 2023*; *Geretz et al., 2023*), integrating these datasets remains difficult due to the varying cell types used in each dataset.

To address these limitations, we conducted single-cell multi-omics analyses on PBMCs from people living with HIV (PLWH) in the early stages of infection. Using a targeted sequencing strategy, we identified and analyzed HIV-1-infected cells at the single-cell level. Based on these findings, we established a gene regulatory network specific to HIV-1-infected cells. This study highlights multiple immune pathways and regulatory factors mediating the effects of acute HIV-1 infection on immune cell function, offering potential therapeutic targets for complete HIV-1 eradication.

## Results

### The single-cell transcriptional landscape of HIV-1-infected cells in early stage of infection

To uncover immune mechanisms and transcriptional regulation in HIV-1-infected cells, we performed single-cell multi-omics sequencing on PBMCs from nine individuals with early-stage HIV infection (<6 months) (*Supplementary file 1A*). Using the 10x Genomics Single-Cell platform, we obtained five scRNA-seq and four snRNA-seq data from single-cell multiome datasets. We integrated scRNA-seq and snRNA-seq data, identifying 16 distinct cell populations based on marker genes (*Figure 1A*; *Figure 1—figure supplement 1A*). We also observed some variability in the proportions of these cell populations among donors, suggesting inter-individual differences in immune cell composition (*Figure 1—figure supplement 1B*). Focusing on CD4+ T cells as main targets of HIV-1, we defined four subtypes: CD4 naive T cells, CD4+ effector memory T cells (CD4 EM cells), T helper (Th)2 cells, and Th17 cells (*Figure 1B*; *Figure 1—figure supplement 1C, D*). *LDHB* and *SELL* were highly expressed in CD4 naive T cells, whereas CD4 EM cells were characterized by high expression levels of *HLA-DRA* and *GZMK*. Among the activated CD4+ T cells, Th2 cells were identified based on the expression of *STAT6* and *GATA3*, and Th17 cells were annotated based on the expression levels of *RORA* and *STAT3*.

To identify HIV-1 RNA+ cells, we aligned scRNA-seq reads to the HIV-1 genome. Additionally, we conducted targeted long-read sequencing using the remaining scRNA-seq libraries to improve the sequencing depth of HIV-1 transcripts. We identified 985 HIV-1 RNA+ cells and 3450 uninfected cells. CD4 naive and EM cells showed a higher proportion of HIV-1 RNA+ cells than Th2 and Th17 (*Figure 1C*; *Figure 1—figure supplement 1E*).

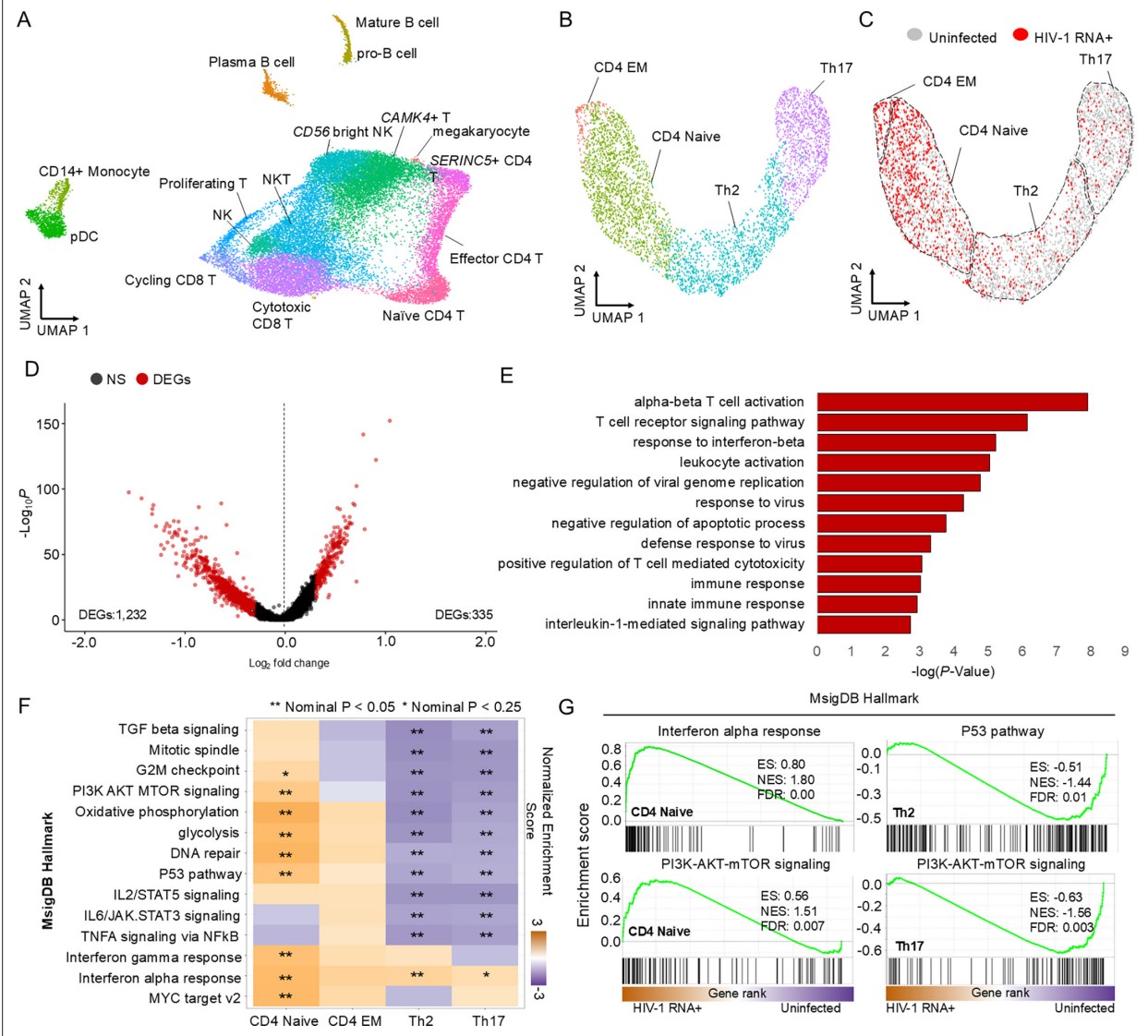

**Figure 1.** Single-cell transcriptomic analysis of HIV-1 RNA+ cells from early infected. (**A**) UMAP plot displays the distribution of PBMCs in early infected patients (*n* = 9), clustered based on transcriptome signatures post-removing batch effects. (**B**) The UMAP plot displays the distribution of 4435 CD4 T cells from early infected patients. (**C**) UMAP of CD4 T cells identifies 3450 uninfected cells and 985 HIV-1 RNA+ cells. (**D**) Volcano plot reveals differentially expressed genes (DEGs) in HIV-1 RNA+ CD4 T cells compared to uninfected cells; red dots signify significant DEGs (adj p-value <0.01). (**E**) Bar plot reveals enriched biological processes and immune pathways in HIV-1 RNA+ CD4 T cells (p-value <0.01) using upregulated DEGs; significance presented as −log(p-value). (**F**) Heatmap exhibits signaling pathways enriched in CD4 T cell types through gene set enrichment analysis (GSEA); positive scores (dark orange) denote enrichment in HIV-1 RNA+ cells (nominal p-value <0.05 and <0.25) are marked with stars. (**G**) GSEA plots show significant immune pathways in CD4 naive, Th2, and Th17 cells, presenting enrichment score (ES), normalized enrichment score (NES), and false discovery rate (FDR).

The online version of this article includes the following figure supplement(s) for figure 1:

**Figure supplement 1.** Single-cell transcriptomic analysis identifies the cell types of PBMCs and subclustered CD4 T cells from early HIV-1-infected patients.

**Figure supplement 2.** Single-cell transcriptomic analysis identifies the characteristics of HIV-1 RNA+ cells.

Gene expression profiling revealed 334 upregulated and 1,002 downregulated genes in HIV-1 RNA+ CD4+ T cells (*Figure 1D*), including genes related to T cell activation (*PRDX2*, *PRDX1*, *HLA-B*, *HLA-C*, *HLA-A*, *HLA-F*, and *HLA-E*), defense responses to viruses (*IFITM3*, *IFITM1*, *IFITM2*, and *MX1*), and interferon-beta responses (*IFITM3*, *BST2*, *IFITM1*, *IFITM2*, *XAF1*, and *SHFL*) (*Figure 1E*). Notably, IFITM1 was the most upregulated gene in HIV-1 RNA+ CD4+ T cells. As an interferon-stimulated

gene, IFITM1 is known to inhibit HIV-1 replication by interfering with viral entry (*Lu et al., 2011*). This finding indicates that the primary cellular pathways active during acute infection are also engaged in the antiviral response to suppress HIV-1.

Given that CD4+ T cell differentiation influences HIV-1 survival (*Horsburgh et al., 2022*), we used gene set enrichment analysis (GSEA) to assess subtype-specific responses (*Figure 1F*). The GSEA results revealed that interferon-gamma and interferon-alpha responses were upregulated in all CD4+ T cell subtypes upon HIV-1 infection. Additionally, hallmarks related to T cell activation were downregulated in both Th2 and Th17 cells. Additionally, while interferon-alpha responses were enriched, downstream pathways (e.g., PI3–AKT–mTOR) were downregulated in Th17 cells (*Figure 1F, G*). Downregulated genes in Th17 cells were linked to PI3K–AKT signaling, while no such enrichment was observed among upregulated genes (*Figure 1—figure supplement 2A, B*). These results suggest that activated T cells experience immune dysfunction in early HIV-1 infection, contrasting with the effects observed in naive CD4+ T cells.

## Characterization of epigenetic changes in HIV-1-infected CD4+ T cells in early infection

Single-cell multiome technology enables a detailed understanding of cell states by simultaneously profiling transcriptome and epigenetic signatures within the same cell, facilitating the construction of an accurate gene regulatory network by identifying gene expression drivers. A combined UMAP was generated based on chromatin accessibility and gene expression, with cell types annotated using marker genes (*Figure 2A*; *Figure 2—figure supplement 1A*). Additionally, coverage plots revealed cell type-specific chromatin accessibility for nearby marker genes (*Figure 2B*). Chromatin accessibility in HIV-1 RNA+ naive and memory CD4+ T cells identified specific patterns, with closed differential accessibility regions (DARs) in naive cells linked to TNF-alpha signaling via NF-κB and open DARs related to HIV-1 Nef protein and signal transduction. In memory CD4+ T cells, closed DARs were related to interferon signaling and negative regulation of viral transcription, whereas open DARs were related to Tat-mediated HIV elongation. When examining genes adjacent to open chromatin regions (*Figure 2D*), we observed high chromatin accessibility for *PSMB7* and *CDK9* in naive and memory CD4+ T cells, respectively, both of which interact with the HIV-1 Tat protein (*Apcher et al., 2003*; *Wang et al., 2020a*). Furthermore, the chromatin accessibility of *NR4A2* and *TRIM62*, which negatively regulate HIV-1 (*Sreeram et al., 2022*; *Chen et al., 2023*; *Uchil et al., 2008*), was reduced in HIV-1 RNA+ cells (*Figure 2E*).

## Identification of key transcription factors governing HIV-1 infection features

Following HIV-1 infection, host transcription factors (TFs) regulate immune responses and viral persistence (*Masenga et al., 2023*). To identify key TFs and construct a gene regulatory network associated with HIV-1 infection, we integrated gene expression and chromatin accessibility data from HIV-1 RNA+ CD4+ T cells.

We found 15 TFs with increased and 24 with decreased regulon activity in HIV-1 RNA+ cells compared to uninfected cells across all CD4+ T cell subtypes (*Figure 3A*). We also confirmed the significantly different regulon activity of these TFs by cell type. CD4 naive and EM cells shared similar TF activity profiles, while Th17 and Th2 cells exhibited more comparable patterns (*Figure 3B*). KLF2, typically high in naive cells and suppressed in activated cells (*Kuo et al., 1997*; *Takada et al., 2011*; *Pedro et al., 2021*), showed elevated regulon activity in naive and EM cells, with lower activity in Th17 and Th2 cells (*Figure 3B*). However, in HIV-1-infected cells, KLF2 activity was upregulated across all subtypes (*Figure 3A*). Conversely, FOXO1 showed higher regulon activity in Th17 and Th2 cells but was downregulated in HIV-1 RNA+ Th17 and Th2 cells. In HIV RNA+ cells, the TFs KLF2 and JUND showed strong regulatory activity and elevated gene expression (*Figure 3C*). Additionally, we identified the upregulated target genes of six TFs (KLF2, JUND, STAT1, JUNB, XBP1, MYC) in HIV-1 RNA+ cells (*Figure 3D*), which were significantly associated with the defense response to viruses, negative regulation of viral genome replication, and innate immune responses (*Figure 2—figure supplement 1B*). As the GSEA result showed different enriched signaling pathways in HIV-1 RNA+ cells by various cell types (*Figure 1G*), several key TF-encoding genes (*KLF2*, *FOXO1*, TBL1*XR1*) which regulate PI3K–ATK signaling and p53 signaling were also differentially expressed in each cell type (*Figure 3E*).

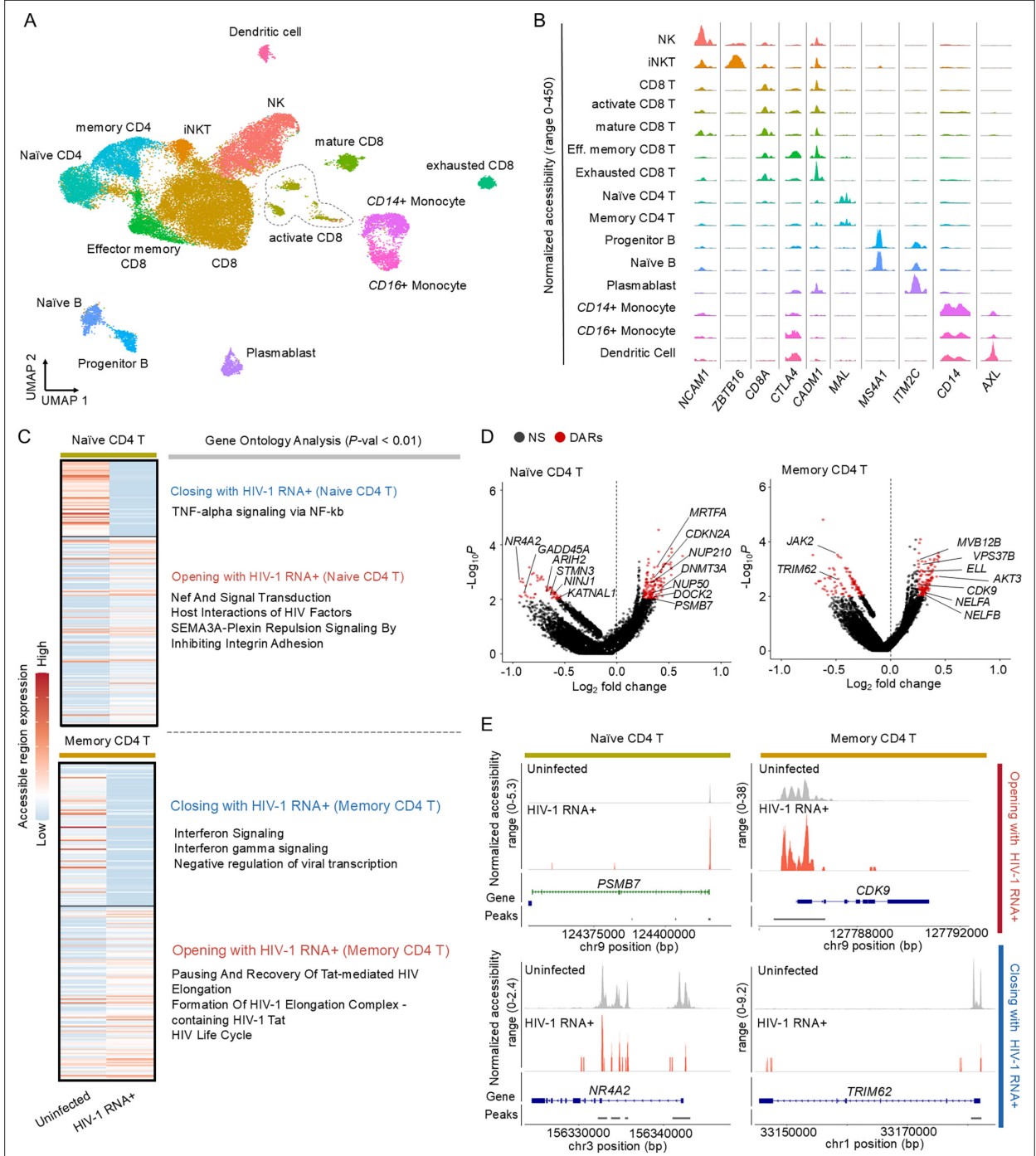

**Figure 2.** Exploring the epigenetic characteristics of HIV-1 RNA+ cells through single-cell multiome data. (**A**) UMAP plot illustrates PBMC distribution in early infected patients (*n* = 8) post-clustering based on transcriptome and epigenome signatures, with batch effects removed. (**B**) Visualization of differential DNA accessibility across 15 cell clusters. Tracks display normalized chromatin accessibility at promoter regions of cluster-specific marker genes. (**C**) Heatmap displays differentially chromatin-accessible regions (DARs) in HIV-1 RNA+ and uninfected cells of naive and memory CD4 T cell types. Right heatmap presents enriched gene ontology for DAR-associated genes. (**D**) Volcano plot shows DARs between HIV-1 RNA+ CD4 T cells and uninfected cells; red dots signify significant DARs, with associated genes displayed. (**E**) Tracks display normalized chromatin accessibility at the promoter regions of key genes (*PSMB7*, *CDK9*, *NR4A2*, *TRIM62*) in HIV-1 RNA+ and uninfected cells.

The online version of this article includes the following figure supplement(s) for figure 2:

**Figure supplement 1.** Single-cell multi-omics analysis identifies epigenetic characteristics of scATAC-seq dataset and upregulated transcription factors (TFs) in HIV-1 RNA+ cells.

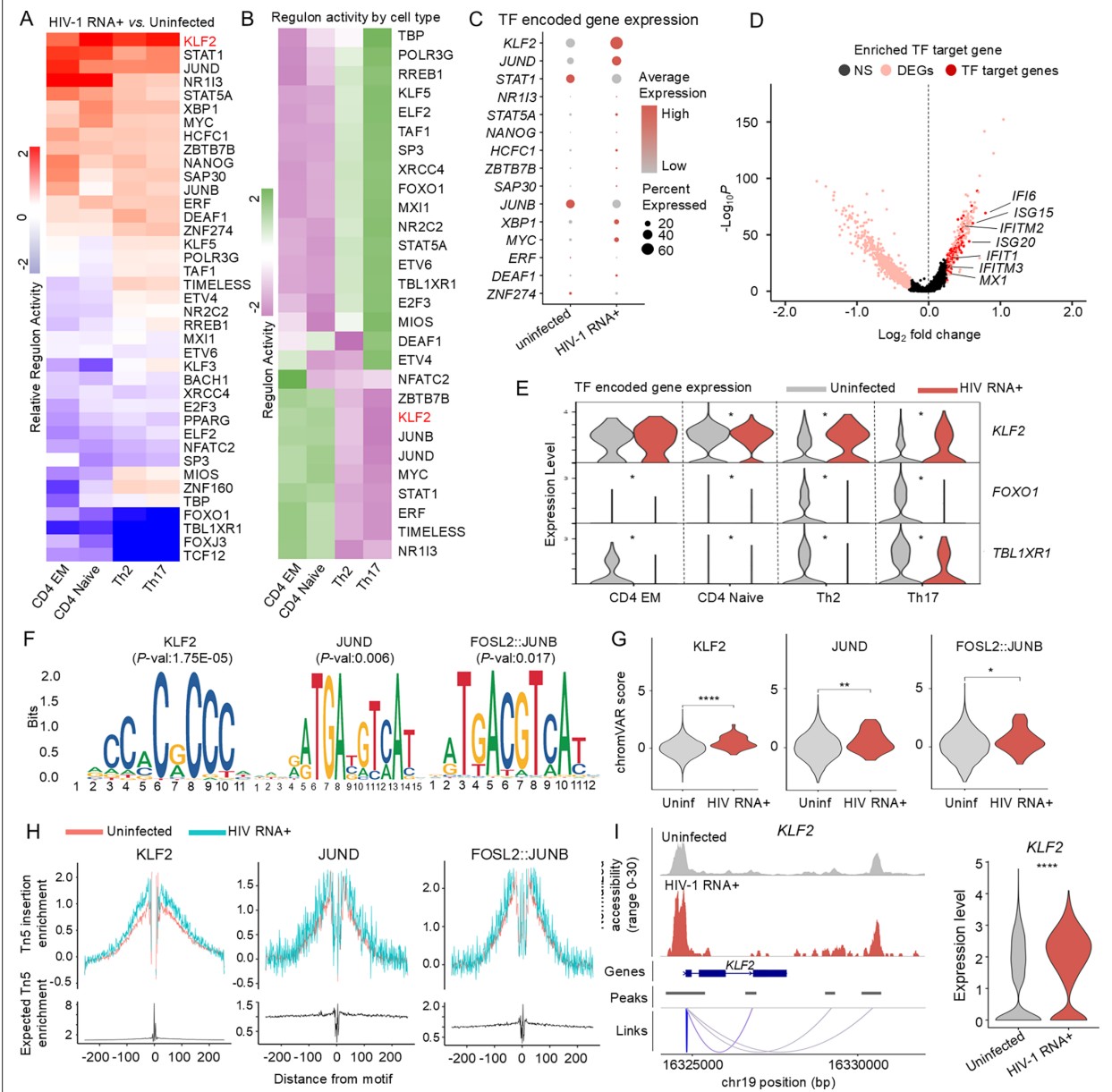

**Figure 3.** Identifying key regulators of HIV-1 RNA+ CD4 T cells through integrated analysis. (**A**) Heatmap depicts relative regulon activity in each CD4 T cell subtype between HIV-1 RNA+ and uninfected cells. Positive values indicate increased activity in HIV-1 RNA+ cells, while negative values represent a decrease. (**B**) Heatmap displays regulon activity levels in each CD4 T cell subtype, with color indicating scaled regulon activity. (**C**) Dot plot displays gene expression of upregulated transcription factors (TFs) in HIV-1 RNA+ cells, with dot size representing the percentage of cells expressing the gene. (**D**) Volcano plot reveals target genes of upregulated TFs in HIV-1 RNA+ cells; pink dots represent DEGs, and red dots indicate target genes of enriched TFs. (**E**) Violin plot illustrates expression of TF-encoding genes in uninfected and HIV-1 RNA+ CD4 T cells across subtypes (*p < 0.01, **p < 0.001, ****p < 0.00001). (**F**) DNA sequences for overrepresented TF-binding motifs in HIV-1 RNA+ T cells compared to uninfected cells. (**G**) Violin plot displays chromVAR motif activity score for enriched motifs. (**H**) TF footprinting profiles show the increased chromatin accessibility near three representative motifs in HIV-1 RNA+ cells. (**I**) Left panel: Tracks display normalized chromatin accessibility at KLF2 gene locus for HIV-1 RNA+ and uninfected cells, with peak-to-gene links bottom of the coverage plot. Right panel: Expression level of KLF2 in each group.

The chromatin accessibility data further supported the increased activity of these TFs in HIV-1 RNA+ cells. We calculated TF activity based on chromatin accessibility and identified enriched TF-binding motifs in HIV-1 RNA+ CD4+ T cells using DARs. KLF2-, JUND-, and FOSL2::JUNB-binding motifs were highly enriched in the DARs opened in HIV-1 RNA+ cells (*Figure 3F*), along with enhanced TF activity (*Figure 3G*). Furthermore, TF footprinting analysis revealed higher chromatin

accessibility adjacent to the KLF2-binding motif sites (*Figure 3H*). The promoter region of *KLF2* was more accessible, with higher mRNA expression in HIV-1 RNA+ cells than in uninfected cells (*Figure 3I*). Furthermore, the putative cis-regulatory element of KLF2 showed more chromatin-accessible peaks in HIV-1 RNA+ CD4+ T cells than in uninfected cells (*Figure 3I*). Overall, this multidimensional approach enabled integrating gene expression and chromatin accessibility in the early stage of infection in PLWH, highlighting potential key TFs that regulate HIV-1 transcription in host cells.

## KLF2 is an essential regulator in the function of HIV-1-infected CD4+ T cells

Using a multidimensional approach, we identified KLF2 as a key TF within the gene regulatory network of HIV-1 RNA+ CD4+ T cells. KLF2 is reported to be involved in the quiescence, survival, and differentiation of T cells (*Hart et al., 2012*). However, its function in viral infection remains poorly understood. To elucidate the role of KLF2, we curated 88 KLF2 target genes from published chromatin immunoprecipitation (ChIP)-chip, ChIP-seq, and other TF-binding site profiling data, as well as SCENIC regulon datasets. Out of 88 KLF2 target genes, 44 were upregulated in HIV-1 RNA+ cells (*Figure 4A*; *Supplementary file 1B*). To validate these findings, we analyzed a publicly available single-cell transcriptomic dataset of HIV-infected PBMCs (*Wang et al., 2020b*). Consistently, expression of *KLF2* and a significant proportion of KLF2 target genes, including *LTB*, *PIM1*, and *NRP2*, were also upregulated in HIV-infected CD4+ T cells (*Figure 4—figure supplement 1A*). These target genes were mainly associated with negative regulation of apoptotic process and negative regulation of programmed cell death (*Figure 4B*). Among these significantly upregulated KLF2 target genes, those reported to be involved in the regulation of HIV-1 replication (*PIM1* and *IL32*) (*Duverger et al., 2014*; *Didichenko et al., 2008*), cell proliferation (*TXNIP*) (*Muri et al., 2021*), and the JAK/STAT pathway (*LTB*) also exhibited higher levels of chromatin accessibility in HIV-1 RNA+ CD4+ T cells than in uninfected cells (*Figure 4C*).

Considering that CD8+ T cells play a critical role in the immune response against HIV-1 infection during chronic infection (*Gulzar and Copeland, 2004*; *Mudd and Lederman, 2014*), we further attempted to identify specific ligand–receptor pairs between CD4+ T cells and cytotoxic T lymphocytes to characterize the immune response induced by HIV-1-infected cells (*Gulzar and Copeland, 2004*). Toward this end, we computed significant ligand–receptor pair interactions between CD8+ and CD4+ T cells using CellPhoneDB (*Figure 4D*; *Efremova et al., 2020*). The results indicated a strong interaction between LTB and its receptor (LTBR) in naive CD4+ T cells, whereas other CD4+ T cell subtypes displayed a weaker interaction with cytotoxic CD8+ T cells. LTB expression was also significantly elevated in HIV-1 RNA+ cells (*Supplementary file 1B*). Additionally, the NRP2 receptor exhibited heightened levels of interaction between CD4+ and CD8+ T cells, particularly in the effector CD4+ T cell population. *NRP2*, a known target gene of KLF2, is expressed on T cells and modulates immune responses under pathological conditions (*Roy et al., 2017*). Given that KLF2 is involved in immune cell trafficking and adhesion, we used CellChat (v2.1.1) (*Jin et al., 2025*) to identify altered ligand–receptor interactions between HIV-1-infected CD4+ T cells and innate immune cells during infection. Both effector and central memory CD4+ T cells exhibited increased MIF and ICAM2 signaling with NK cells, which are associated with KLF2-mediated immune modulation and may enhance immune activation by facilitating immune cell adhesion and migration (*Figure 4—figure supplement 2A*; *Pestal et al., 2024*; *SenBanerjee et al., 2004*). However, HIV-1-infected CD4+ T cells simultaneously showed reduced CCL5 and CLEC2B signaling, potentially limited monocyte-driven responses and NK cell recruitment (*Gouwy et al., 2011*; *Welte et al., 2006*). These changes in immune interaction may represent a balance between immune activation and evasion, contributing to viral persistence.

Subsequently, we explored the interactions between the host proteins encoded by KLF2 target genes and HIV-1 proteins. Using the STRING database, we identified 16 KLF2 target genes that encode proteins that directly interact with HIV-1 (*Figure 4E*). One of these target genes, *SOCS3*, which encodes a protein exhibiting a strong interaction with the Tat protein, pivotal for viral genome replication (*Akhtar et al., 2010*). Another KLF2 target gene, *PIM1*, which is known as a transactivator of Tat (*Didichenko et al., 2008*), displayed substantial interaction with JUNB and exhibited high gene expression and chromatin accessibility in HIV-1 RNA+ cells (*Figure 4C*).

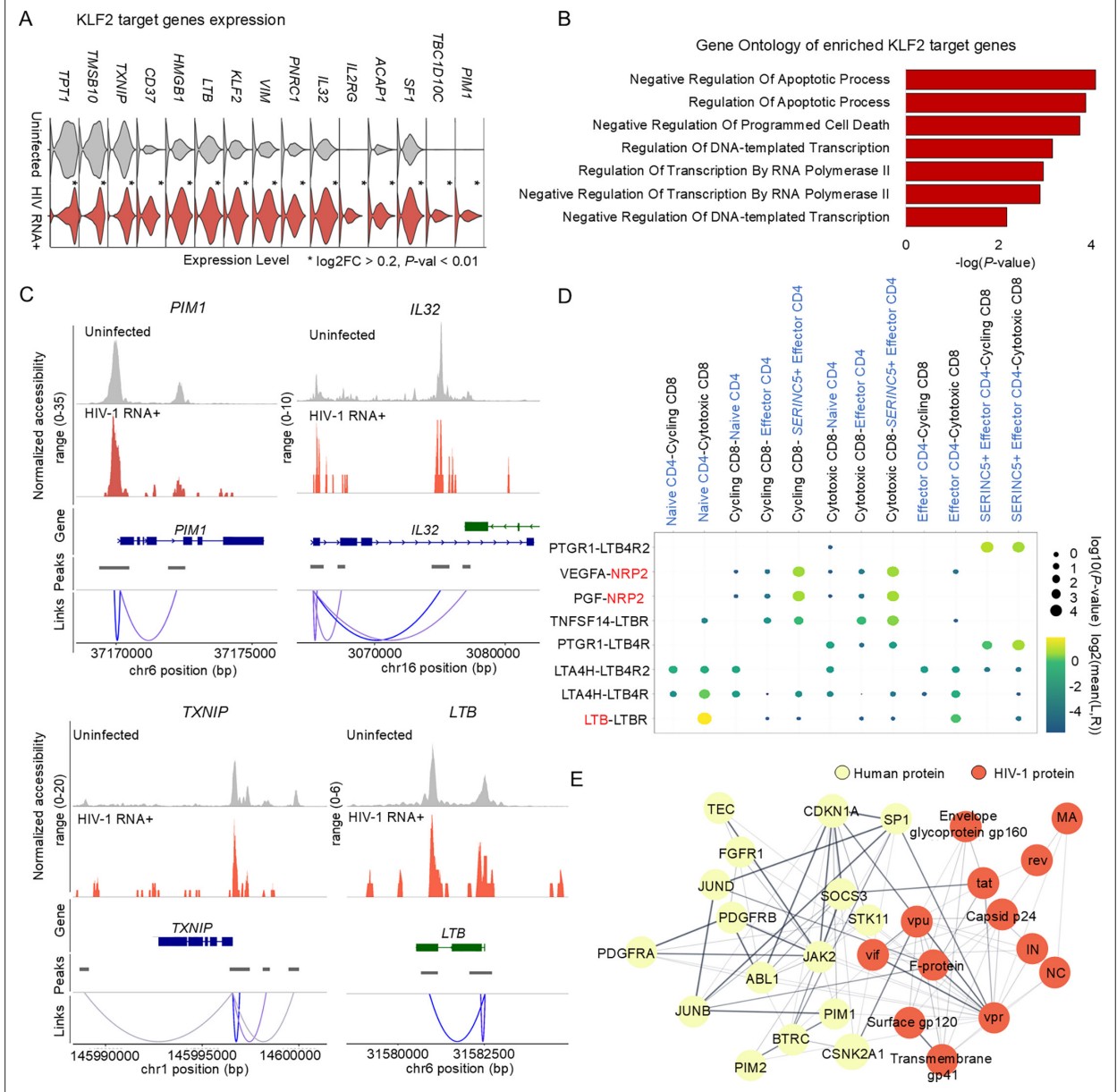

**Figure 4.** Functional insights into KLF2 upregulation in HIV-1 RNA+ CD4 T cells. (**A**) Bar plot displays gene ontologies associated with upregulated KLF2 transcription factor (TF) target genes in HIV-1 RNA+ cells (p-value <0.01; significance expressed as −log(p-value)). (**B**) Violin plot represents expression of upregulated KLF2 TF target genes in uninfected cells and HIV-1 RNA+ CD4 T cells (*p < 0.01). (**C**) Tracks display normalized chromatin accessibility at promoter loci of *PIM1*, *IL32*, *TXNIP*, and *LTB* for HIV-1 RNA+ and uninfected cells, with peak-to-gene links bottom of the coverage plot. (**D**) Dot plot displays ligand–receptor pairs detected between CD4 and CD8 T cells; red-labeled gene symbols are KLF2 target genes. Circle size indicates p-values, while color represents mean average expression levels. (**E**) Interactions between KLF2 target genes and HIV-1 proteins identified using STRING database. Yellow dots represent human proteins, and red dots represent HIV-1 proteins; line thickness indicates interaction strength.

The online version of this article includes the following figure supplement(s) for figure 4:

**Figure supplement 1.** KLF2 target gene expression in HIV-1-infected CD4+ T cells.

**Figure supplement 2.** Cell–cell interaction in HIV-1-infected CD4+ T cells.

## Th17 cells exhibit increased enrichment and susceptibility to HIV-1 infection

In vivo analysis of the states of cells with the HIV-1 provirus integrated into host genome has been a long-standing challenge in HIV research. To address this issue, we defined cells with two or more

scATAC-seq reads that aligned to the HIV-1 genome as HIV-1 DNA+ cells. The sequencing reads predominantly aligned to the 5'- and 3'-long terminal repeat (LTR) regions of the HIV-1 genome (*Figure 5A*). We observed clustered HIV-1 DNA+ cells in a distinct region, indicating that these cells share a similar epigenetic profile (*Figure 5B*; *Figure 5—figure supplement 1A*). DARs were identified in the cluster enriched with HIV-1 DNA+ CD4+ T cells compared to the remaining CD4+ T cells (*Figure 5C*). The genes closest to the DARs that showed increased accessibility levels in the HIV-1 DNA+ CD4+ T cell cluster compared to the corresponding levels in the uninfected clusters were associated with the inflammatory and defense responses, which was in line with the activation of the innate immune response observed in HIV-1 RNA+ cells (*Figure 5D*).

Notably, we identified distinctly enriched TF-binding motifs in HIV-1 DNA+ CD4+ T cells and HIV-1 RNA+ CD4+ T cells. The binding motifs for key regulators of Th17 cell differentiation (RORC, STAT3, and RORA) were enriched in HIV-1 DNA+ CD4+ T cells (*Figure 5E, F*; *Castro et al., 2017*). These TFs were also reported to regulate viral gene expression by binding to the HIV-1 LTRs (*Wiche Salinas et al., 2021*). Overall, HIV-1 DNA+ cells mostly represented the Th17 cell type (*Figure 5—figure supplement 1B, C*), in contrast to HIV-1 RNA+ CD4+ T cells, which belonged to various subtypes. Additionally, we confirmed significant enrichment of the KLF2 TF-binding motif in HIV-1 DNA+ CD4+ T cells, as found for HIV-1 RNA+ CD4+ T cells. Furthermore, we detected increased gene expression levels and chromatin accessibility of *CCR6*, *IL4I1*, *NFKB1*, and *CCR5* (*Figure 5G*).

## Discussion

In response to early HIV-1 infection, various host immune responses are triggered, contingent upon the subtype of CD4+ T cells. Our study utilized scRNA-seq and scATAC-seq datasets to profile PBMCs from early HIV-1-infected patients, revealing specific immune responses and mechanisms in distinct HIV-1-infected CD4+ T cell subsets that contribute to HIV pathogenesis. We further identified key regulators of HIV-1 persistence in CD4+ T cells, offering new targets for complete viral elimination.

Interferons play a crucial role in the initial antiviral response during early viral infection by upregulating interferon-stimulated genes and activating signaling pathways like JAK/STAT-, PI3K-, and p53-dependent pathways. These pathways facilitate T cell differentiation and selective proliferation of antiviral T cells, effectively inhibiting viral replication and clearing infected cells (*Satarker et al., 2021*). However, our findings indicate that during acute HIV-1 infection, Th2 and Th17 cells exhibit heightened interferon-alpha responses, despite diminished downstream signaling pathways. In parallel, we observed a higher proportion of HIV-1-infected naive CD4+ T cells. Although their baseline transcriptional activity is low, previous studies have shown that naive CD4 T cells are susceptible to HIV-1 infection during the early stage of infection, potentially due to dysfunctional antiviral signaling in resting cells (*Douek et al., 2002*; *Jiao et al., 2014*). These findings suggest a potential immunological malfunction induced by HIV-1, hindering host antiviral responses by disrupting interferon target signaling, akin to strategies observed in other viruses like Influenza, HCV, and HSV (*Katze et al., 2002*).

In CD4 memory T cells, we observed enriched chromatin accessibility linked to Tat-mediated HIV-1 elongation, concurrent with downregulated interferon-gamma signaling. This underscores HIV-1's ability to persist in memory CD4+ T cells and evade host antiviral responses early in infection. Through integrated analysis, we identified KLF2 as a pivotal TF within the regulatory network of HIV-1 RNA+ CD4+ T cells. KLF2, known for promoting cellular quiescence and regulating T cell migration, exhibited sustained expression across all HIV-1-infected CD4+ T cell subtypes, including activated cells (*Kuo et al., 1997*). Upregulated KLF2 target genes associated with HIV-1 replication and TXNIP, a cell proliferation inhibitor, further suggest HIV-1's ability to persist without triggering transcriptional repressors or proliferation. These findings were further supported by analysis of a publicly available scRNA-seq dataset from HIV-1-infected PBMCs, in which a similar upregulation of *KLF2* and KLF2 target genes was observed in HIV-1-infected CD4+ T cells, reinforcing the reproducibility of our results. A recent study identified FOXP1 and GATA3 as transcriptional regulators of HIV-1 RNA expression using both cell line-based and primary CD4+ T cell models (*Ashokkumar et al., 2024*). While these findings highlight transcriptional control in HIV-1 latency, we did not observe notable differential expression of FOXP1 or GATA3 in our dataset (*Figure 4—figure supplement 1B*). This discrepancy may reflect differences in infection models, stages, or cellular contexts. Nonetheless, both studies underscore the importance of transcriptional regulation in sustaining HIV-1 persistence.

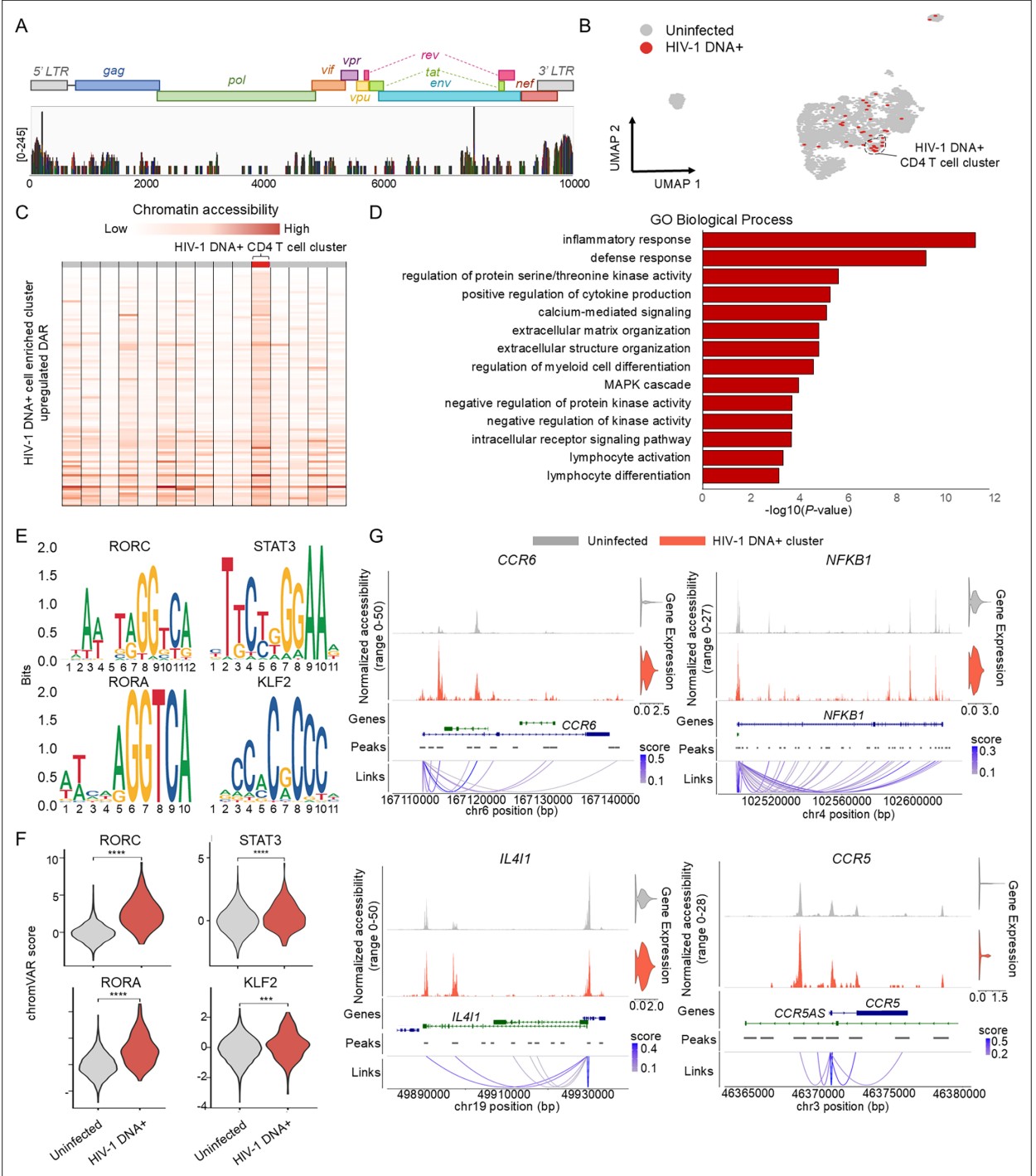

**Figure 5.** Epigenetic analysis in HIV-1 DNA+ CD4 T cells. (**A**) Bar plot presents scATAC-seq reads aligned to the HIV-1 genome, showing coverage for individual genomic windows (log-scaled, max value: 245). (**B**) UMAP of CD4 T cells based on chromatin accessibility identifies 5535 uninfected cells and 39 HIV-1 DNA+ cells. (**C**) Heatmap displays chromatin accessibility levels of differential accessibility regions (DARs) enriched in HIV-1 DNA+ CD4 T cell cluster. (**D**) Bar plot represents gene ontologies associated with genes near DARs of HIV-1 DNA+ CD4 T cell cluster; significance expressed as −log(p-value). (**E**) DNA sequences for overrepresented transcription factor-binding motifs in HIV-1 DNA+ CD4 T cell cluster compared to the rest of CD4 T cells. (**F**) Violin plot displays chromVAR motif activity score for enriched motifs. (***p < 0.0001, ****p < 0.00001) (**G**) Bar plots on the left show normalized ATAC reads for individual genes in HIV-1 DNA+ CD4 T cells and uninfected cells. Bottom shows peak-to-gene links; loop color strength signifies significance of links between peak and gene promoter. The right violin plot displays gene expression levels.

The online version of this article includes the following figure supplement(s) for figure 5:

**Figure supplement 1.** Single-cell epigenetic analysis identifies HIV-1 DNA+ cells are mostly represented by Th17 cell type.

Using scATAC-seq data, we differentiated HIV-1 DNA+ CD4+ T cells, revealing enriched TF motifs (RORC, STAT3, RORA) indicative of Th17 cell predominance. These cells, unlike HIV-1 RNA+ CD4+ T cells distributed across various subtypes, exhibited heightened susceptibility and integration of HIV DNA, with increased expression of *CCR6* and *CCR5*—critical HIV-1 co-receptors (*Richardson et al., 2012*; *Hütter et al., 2009*). Moreover, enriched KLF2 motifs in HIV-1 DNA+ cells positively regulate *CCR5* expression, along with increased *IL4I1* expression linked to Th17 cell immunosuppression and *NFKB1* involved in HIV-1 genome transcription initiation (*Cousin et al., 2015*; *Coiras et al., 2007*). This suggests Th17 cells' potential role as reservoirs due to their susceptibility, quiescent nature, and stem cell-like characteristics.

Limitations include the inability of our HIV-1 targeted sequencing strategy to distinguish HIV-1 RNA from DNA within single cells, necessitating deeper sequencing to capture transcriptionally silenced proviruses. Our study's design focused on early infection stages (<6 months), limiting insights into longitudinal changes in cell populations and gene expression profiles during HIV-1 progression. Given the heterogeneity of HIV-1 infection among PLWH, our findings primarily apply to early infection stages and require validation across different infection phases.

In summary, our study unveils disrupted immune mechanisms in distinct HIV-1 RNA+ CD4+ T cell subsets and underscores KLF2's role in HIV-1-infected cells. We propose that this dysregulation enables HIV-1 transcription maintenance while attenuating pathogenic immune responses. Furthermore, Th17 cells may serve as reservoirs due to their heightened susceptibility and KLF2-mediated CCR5 expression. Our findings provide insights into the complex transcriptional and epigenetic changes during HIV-1 infection, informing potential immunomodulatory therapies to restore normal immune responses and enhance viral control in PLWH, thereby limiting disease progression.

## Methods

### Preparation of PBMC samples from early infection of PLWH

PBMCs were collected from a total of nine individuals in the early stages of HIV infection (<6 months). Epidemiological and virological characteristics of these participants at baseline, including sex, age, duration from HIV-1 infection, CD4 T cell count, HIV-1 viral load, and HIV-1 p24 antigen/antibody screening (*Supplementary file 1A*).

### Preparation of scRNA-seq libraries

The libraries were prepared using the Chromium Single Cell 3′ Library & Gel Bead Kit v2 (PN-120237) (10x Genomics) following the manufacturer's instructions. The single-cell suspension was washed twice with 1x PBS containing 0.04% BSA. Cell number and viability were determined using the Bio-Rad TC20 cell counter. The cells were loaded onto the 10x Genomics Chromium Controller to generate gel beads in emulsions (GEMs). Library preparation was performed according to the 10× Genomics Chromium Single Cell 3′ reagent kit (V2 chemistry) instructions. The quality and concentration of the libraries were assessed using the Agilent Bioanalyzer 2100. Finally, the libraries were sequenced on an Illumina NovaSeq.

### Preparation of scATAC-seq libraries

The libraries were prepared using the Chromium Next GEM Single Cell ATAC Library & Gel Bead Kit (PN-1000175) following the manufacturer's instructions. After isolating the nuclei from PBMCs, the single-nuclei suspension was mixed with Transposition Mix and incubated for 60 min at 37°C. Subsequently, the transposed nuclei were combined with the master mix and loaded onto the 10x Genomics Chromium Controller to generate gel beads in emulsions (GEMs). Library construction was performed according to the instructions provided with the Chromium Next GEM Single Cell ATAC Reagent Kits (v1.1). The subsequent steps were performed in the same way as described in 'Preparation of scRNA-seq libraries'.

### Preparation of single-cell multiome libraries

The libraries were prepared using the Chromium Next GEM Single Cell Multiome ATAC + Gene Expression Reagent Kits (PN-1000283) (10x Genomics) following the manufacturer's instructions. Briefly, nuclei were mixed with Transposition Mix and incubated for 60 min at 37°C. Then these were

loaded onto Chromium Next GEM Chip J (10x Genomics) and GEMs were generated according to the manufacturer's instructions. After post-GEM cleaned up, barcoded transposed DNA and barcoded full-length cDNA fragments were pre-amplified with PCR. ATAC libraries were constructed using the Library Construction Kit (PN-1000190) and full-length pre-amplified cDNA fragments were additionally amplified via PCR. RNA libraries were constructed using the Library Construction Kit B (PN-1000279). Finally, the libraries were sequenced on an Illumina NovaSeq 6000.

### HIV-1 targeted sequencing

After scRNA-seq, remaining scRNA-seq full-length cDNA libraries were subjected to HIV-1 targeted sequencing using the SQK-LSK-109 Ligation Sequencing Kit (Oxford Nanopore Technologies) and three specific primers for HIV-1 (*Supplementary file 1C*). The TruSeq Read1 primer was used to preserve the cell barcode information, and the cDNA fragments containing the HIV-1 sequence were amplified by the HIV-1-specific primers. The cDNA fragments with HIV-1 were primarily selected through PCR using biotinylated primers. Subsequently, the HIV-1-specific primers selectively amplified the cDNA fragments through hemi-nested PCR. These selected cDNA fragments were sequenced on a MinION FLO-MIN106 R9.4.1 flow cell (Oxford Nanopore Technologies). Base calling was performed by using Guppy (V6.1.1), and cells with HIV-1 reads were identified using the cell barcode information.

### Sequence alignment to human and HIV-1 genome

The reads from scRNA-seq and scATAC-seq were aligned to GRCh38 human genome (v.3.0.0) using CellRanger count (10x Genomics, v.6.1.2) and CellRanger ARC (v.2.0.0) with default options, respectively. Feature-barcode matrices were used for downstream analysis. To identify the sequencing reads containing the HIV-1 sequence, the reads from scRNA-seq were aligned to the HIV-1 genome using the STARsolo (v. 2.7.1a). We adjusted the mapping percentage parameter from 50% to 25% to map the short length of HIV-1 sequence included in the sequencing reads.

### scRNA-seq data processing

For the scRNA-seq datasets, Seurat R package (v.3.2.1) was used for processing the raw count matrices (UMI counts per gene per cell) *Stuart et al., 2019*. Prior to feature filtering, DoubletFinder was used to identify and remove putative doublets (*McGinnis et al., 2019*). As in previous studies, standard quality control and preprocessing steps were performed for downstream analysis (*Kim et al., 2024a*; *Kim et al., 2024b*). Cells with fewer than 200 features and greater than 30% mitochondrial gene expression were excluded, and cells expressing fewer than three genes were eliminated. To ensure consistency in quality control and analysis, published scRNA-seq datasets were processed using the same Seurat-based pipeline as described above. Subsequently, the dataset was normalized, highly variable genes were selected, and scaling was performed in Seurat. Batch effects were removed using the 'RunFastMNN' function, and principal component analysis was performed after integrating the datasets. Cells were clustered based on the computed nearest neighbor map using the 'FindNeighbors' function. The clustered cells were visualized on UMAP embeddings. Differentially expressed genes in each cluster were identified using the 'FindAllMarkers' function, and the clusters were annotated using the known cell markers. Clusters were merged if the number of differentially expressed genes was less than 10.

### scATAC-seq data processing

For the scATAC-seq datasets, Signac R package (v.1.5.0) was used to process the data (*Stuart et al., 2021*). To ensure common features across the datasets, combined peaks were generated. The transcription start site (TSS) enrichment score and nucleosome signal score were calculated for each cell using the 'TSSEnrichment' and 'NucleosomeSignal' functions in Signac. Cells were retained with a TSS enrichment score >1 and a nucleosome signal <2 and further filtered the cells containing <1000 or >100,000 ATAC fragments. MACS2 was utilized to identify peaks using each fragment file of datasets (*Zhang et al., 2008*). The datasets were merged, and batch effects were removed using Harmony (*Korsunsky et al., 2019*). The merged dataset was subjected to dimensional reduction with LSI using the 'RunTFIDF', 'FindTopFeatures', and 'RunSVD' functions. The LSI components were used for UMAP projection. The cell type of each cluster was annotated using differentially accessible peaks. To identify significant peaks for each cluster, the 'FindAllMarks()'

function was employed with the following parameters: min.pct=0.1, test.use = 'LR', only.pos= TRUE. The 'ClosestFeature()' function was then used to find the closest genes to each peak within each cluster. Cell clusters were annotated using known cell markers. Subsequently, the scRNA-seq dataset and scATAC dataset from the 10x multiome dataset were integrated using identical cell barcodes to ensure the scATAC-seq clusters' cell type annotation, leveraging the previously annotated scRNA-seq cell types.

## Gene ontology enrichment analysis and GSEA

To identify enriched biological pathways in HIV-1 RNA+ CD4 T cells, DEGs for each cell type were identified using 'FindMarkers' function in Seurat (v.3.2.1). We compared the normalized gene expression data of HIV-1 RNA+ CD4 T cells with uninfected CD4 T cells. All genes that passed quality control were included in the DEG analysis without additional filtering.

For gene ontology enrichment analysis, DEGs were filtered based on |logFC| >0.25 and adjusted p-value <0.01. Gene ontology enrichment analysis was performed using the web-based tools of DAVID (version DAVID 6.8; http://david.ncifcrf.gov/; RRID:SCR_001881). Biological processes (BPs) were ranked based on the sets of DEGs in HIV-1 RNA+ cell, considering BPs with a p-value <0.01 significant.

For GSEA, the raw count matrix was used, and no additional filtering was applied. GSEA was performed to identify the statistically significant gene sets in the HIV-1 RNA+ cell group within different CD4 T cell subtypes. Hallmark gene sets from Molecular Signatures Database (MSigDB) were used, and only gene sets containing 10–500 genes were selected.

## Gene regulatory network analysis

To construct a gene regulatory network and predict TF activities from a gene expression dataset, SCENIC (R package, https://github.com/aertslab/SCENIC; RRID:SCR_017247) was utilized as a computational method (*Aibar et al., 2017*). Co-expression modules between TFs and target genes were identified using GENIE3. The cisTarget Human motif database was utilized for scoring the regulons. Subsequently, the AUCell method was employed to score single cells and identify enrichment of regulons in the infection group.

## Cell–cell interaction analysis

CellphoneDB was utilized to analyze cell–cell communication between CD4 T cell subtypes and CD8 T cell subtypes (*Efremova et al., 2020*). Interaction pairs with a positive mean expression level in the HIV-1 RNA+ group were selected to explore associations among cell types. The interaction strength was visualized using the 'plot_cpdb' function in the ktplots R package, represented as a dot plot. To identify dysregulated ligand–receptor signaling, we additionally applied CellChat (v2.1.1) (*Jin et al., 2025*). Differential expression analysis was performed between HIV-1 RNA+ and uninfected CD4+ T cells using the 'identifyOverExpressedGenes' function. Upregulated and downregulated interactions were determined based on fold-change values.

## TF-binding motif enrichment and footprinting analysis

The cells were categorized into three groups: HIV-1 RNA+ cell group, HIV-1 DNA+ cell group, and uninfected cell group. DARs among the groups were identified using the 'FindMarkers' function. DARs with a p-value less than 0.01 between the cell groups were filtered. DARs with a positive $\log_2$ fold-change value were selected for further analysis to identify enriched motifs. Enriched motifs were identified using the motifmatchr (R package, https://bioconductor.org/packages/motifmatchr; RRID:SCR_026739), which involved testing for overrepresentation of each DNA sequence motif in the DARs using a hypergeometric test. The computation of GC contents and matching of background sets of peaks for GC content, sequence length, and counts were performed using the 'FindMotifs' function in Signac. The per-cell motif activity score was calculated by chromVAR for each motif across the infection conditions (*Schep et al., 2017*). For the identification of a strong enrichment of Tn5 integration events adjacent to the enriched motifs, TF footprinting analysis was conducted using the 'Footprint' function in Signac.

## Statistical analysis

The subsection above provides a description of statistical analyses for single-cell transcriptomic and epigenetic studies. All descriptive statistical analyses were conducted in R 3.6.0 and higher. Benjamini–Hochberg correction was used for significance.

## Acknowledgements

We thank all study participants for donating their blood and making this study possible. We are grateful to Jaehyun Seong for preparing this study samples of acute HIV-infected patients. The study samples of acute HIV-infected patients are collected by the Chronic Infectious Disease Cohort Study (Grant Number: 4800-4859-304). This work was supported by the Korea National Institute of Health (Grant Number: 2019-NI-067 and 2020-ER5103), the National Research Foundation of Korea (NRF), funded by the Korean government (Grant Number: RS-2024-00335026), and a grant of the Korea-US Collaborative Research Fund (KUCRF), funded by the Ministry of Science and ICT and Ministry of Health & Welfare, Republic of Korea (Grant Number: RS-2024-00466906).

## Additional information

### Competing interests

Jihwan Park: Reviewing editor, eLife. The other authors declare that no competing interests exist.

### Funding

| Funder | Grant reference number | Author |
|---|---|---|
| Korea National Institute of Health | 2019-NI-067 | Byeong-Sun Choi Jihwan Park |
| Korea National Institute of Health | 2020-ER5103 | Byeong-Sun Choi Jihwan Park |
| National Research Foundation of Korea | RS-2024-00335026 | Byeong-Sun Choi Jihwan Park |
| Ministry of Science and ICT, South Korea | RS-2024-00466906 | Byeong-Sun Choi Jihwan Park |
| Ministry of Health and Welfare | RS-2024-00466906 | Byeong-Sun Choi Jihwan Park |

The funders had no role in study design, data collection, and interpretation, or the decision to submit the work for publication.

### Author contributions

Dayeon Lee, Resources, Data curation, Formal analysis, Visualization, Methodology, Writing – original draft, Writing – review and editing; Sin Young Choi, So-I Shin, Resources, Investigation; Hyunsu An, Resources, Investigation, Methodology; Byeong-Sun Choi, Conceptualization, Resources, Supervision, Funding acquisition, Project administration, Writing – review and editing; Jihwan Park, Conceptualization, Supervision, Funding acquisition, Project administration, Writing – review and editing

### Author ORCIDs

Dayeon Lee ⬩ https://orcid.org/0000-0002-8006-8704
Byeong-Sun Choi ⬩ https://orcid.org/0000-0002-6128-7995
Jihwan Park ⬩ https://orcid.org/0000-0002-5728-912X

### Ethics

Written informed consent was obtained from all participants. The study was approved by the Korea Disease Control and Prevention Agency (IRB No. 2019-07-06-P-A) and conducted in accordance with the principles of the Declaration of Helsinki.

Reviewer #2 (Public review): https://doi.org/10.7554/eLife.104856.3.sa1

Reviewer #3 (Public review): https://doi.org/10.7554/eLife.104856.3.sa2
Author response https://doi.org/10.7554/eLife.104856.3.sa3

## Additional files

### Supplementary files

Supplementary file 1. Patient characteristics, KLF2-associated DEGs in CD4 T cells, and primers for HIV-1 targeted sequencing. (**A**) Epidemiological and virological characteristics, and sequencing data information for nine acute HIV-infected patients at baseline. (**B**) Differentially expressed genes overlapped with KLF2 target gene in CD4 T cell (HIV-1 RNA+ vs. uninfected). (**C**) Primers used for HIV-1 targeted sequencing.

Supplementary file 2. Table of differentially accessible regions between HIV RNA+ CD4 naïve T cells and uninfected cells in ATAC-seq data.

Supplementary file 3. Table of differentially accessible regions between HIV RNA+ CD4 central memory T cells and uninfected cells in ATAC-seq data.

Supplementary file 4. Table of differentially accessible regions between HIV DNA+ CD4 T cells and uninfected cells in ATAC-seq data.

MDAR checklist

### Data availability

The raw sequencing data (accession number: KAP230707) have been deposited in the Korea National Research Data Archive (KRA) but are not publicly available due to ethical and institutional restrictions associated with human biospecimens. Interested researchers may request access by submitting a data use application via the KRA system. The application, which includes information about the applicant and the proposed research project, will be reviewed and approved by the data depositor (corresponding author: jihwan.park@gist.ac.kr) in accordance with institutional policies. Requests are typically reviewed within approximately 2–4 weeks. Data will be provided for non-commercial academic research purposes only. All software used for data analysis is described in the Methods section. Processed summary data, including differential expression gene (DEG) tables and differential accessibility region (DAR) tables used to generate figures and tables in the manuscript, are provided in *Supplementary files 1–4*.

The following dataset was generated:

| Author(s) | Year | Dataset title | Dataset URL | Database and Identifier |
|---|---|---|---|---|
| Park J | 2024 | Study on people living with early HIV-1 infection using multimodal single-cell sequencing analysis | https://kbds.re.kr/KAP230707 | Korea BioData Station, KAP230707 |

The following previously published datasets were used:

| Author(s) | Year | Dataset title | Dataset URL | Database and Identifier |
|---|---|---|---|---|
| Wang S, Zhang Q, Hui H | 2020 | An Atlas of Immune Cell Exhaustion in HIV-Infected Individuals Revealed by Single-Cell Transcriptomics | https://www.ncbi.nlm.nih.gov/geo/query/acc.cgi?acc=GSE157829 | NCBI Gene Expression Omnibus, GSE157829 |
| Ashokkumar M, Mei W, Peterson JJ, Margolis D, Jiang Y, Browne E | 2023 | Integrated single-cell multiomic analysis of HIV latency reversal reveals novel regulators of viral reactivation | https://www.ncbi.nlm.nih.gov/geo/query/acc.cgi?acc=GSE242997 | NCBI Gene Expression Omnibus, GSE242997 |

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
