## [Editor Report · eLife Assessment]

This study presents **important** findings that enhance our understanding of immune cell interactions in the context of chronic HIV-1 infection. The evidence supporting the conclusions is **convincing**. The authors have employed appropriate and validated methodologies, including detailed data reprocessing and batch correction to account for inter-donor variability. The inclusion of supplementary figures and analyses, such as cell communication inference, further substantiates the robustness of the findings. Overall, this work contributes to our understanding of HIV-1 immune evasion and highlights potential therapeutic targets for reservoir eradication.

---

## [Referee Report · Reviewer #2 (Public review)]

Summary:

The authors observed gene ontologies associated with upregulated KLF2 target genes in HIV-1 RNA+ CD4 T Cells using scRNA-seq and scATAC-seq datasets from the PBMCs of early HIV-1-infected patients, showing immune responses contributing to HIV pathogenesis and novel targets for viral elimination.

Strengths:

The authors carried out detailed transcriptomics profiling with scRNA-seq and scATAC-seq datasets to conclude upregulated KLF2 target genes in HIV-1 RNA+ CD4 T Cells.

Comments on revisions:

The authors justified my comments.

---

## [Referee Report · Reviewer #3 (Public review)]

The revised manuscript demonstrates a marked improvement over the previous version. The authors have successfully incorporated feedback, and have moreover expanded their analyses.

The Methods section is now more detailed and meets the requirements for reproducible research. Authors have reprocessed the data, creating an integrated dataset using a previously published single-cell RNA-Seq atlas, which includes both healthy donors and individuals with chronic HIV-1 infection. An additional batch correction step was included into the processing pipeline after the explicit analysis of inter-donor variability within immune subsets, as was suggested.

Several supplementary figures were added, which both improve the understanding of data and address questions raised by the reviewers. The manuscript also provides additional analysis of cell communication inference, as suggested. The study of interactions between NK cells and infected CD4+ T cells, as well as between monocytes and infected CD4+ T cells, is valuable for understanding the influence of cell signaling on antiviral response and the production of HIV-1 transcripts in infected cells.

The authors have addressed all the reviewers' suggestions, and the current version of the manuscript is both more comprehensive and more informative. Additional analysis has strengthened the narrative and the reproducibility of the research.

The resulting manuscript is both more robust and more informative.

---

## [Author Response]

The following is the authors’ response to the original reviews.

**Public Reviews:**

**Reviewer #1 (Public review):**
Summary:The authors aimed to elucidate the molecular mechanisms underlying HIV-1 persistence and host immune dysfunction in CD4+ T cells during early infection (<6 months). Using single-cell multi-omics technologies-including scRNA-seq, scATAC-seq, and single-cell multiome analyses-they characterized the transcriptional and epigenomic landscapes of HIV-1-infected CD4+ T cells. They identified key transcription factors (TFs), signaling pathways, and T cell subtypes involved in HIV-1 persistence, particularly highlighting KLF2 and Th17 cells as critical regulators of immune suppression. The study provides new insights into immune dysregulation during early HIV-1 infection and reveals potential epigenetic regulatory mechanisms in HIV-1-infected T cells.Strengths:The study excels through its innovative integration of single-cell multi-omics technologies, enabling detailed analysis of gene regulatory networks in HIV-1-infected cells. Focusing on early infection stages, it fills a crucial knowledge gap in understanding initial immune responses and viral reservoir establishment. The identification of KLF2 as a key transcription factor and Th17 cells as major viral reservoirs, supported by comprehensive bioinformatics analyses, provides robust evidence for the study's conclusions. These findings have immediate clinical relevance by identifying potential therapeutic targets for HIV-1 reservoir eradication.

We sincerely appreciate the reviewer’s positive evaluation of our work.

Weaknesses:Despite its strengths, the study has several limitations. By focusing exclusively on CD4+ T cells, the study overlooks other relevant immune cells such as CD14+ monocytes, NK cells, and B cells. Additionally, while the authors generated their own single-cell datasets, they need to validate their findings using other publicly available single-cell data from HIV-1-infected PBMCs.

Thank you to Reviewer #1 for your feedback on our work. In response to this feedback, we have examined cell-cell interactions between HIV-1-infected CD4+ T cells and other innate immune cells, including monocytes and NK cells. We identified altered interaction signaling patterns (e.g., MIF, ICAM2, CCL5, CLEC2B) that contribute to immune dysfunction and viral persistence (page 9, Supplementary Fig. 5)In addition, we validated the expression of KLF2 and its target genes using a publicly available scRNA-seq dataset from HIV-1-infected PBMCs [1], which includes both healthy donors and individuals with chronic HIV-1 infection. The upregulation of key KLF2 targets in HIV-1-infected CD4+ T cells from this dataset supports the reproducibility of our findings. We have incorporated into the revised Results, Discussion, and Supplementary Materials (page 8, page 12 and Supplementary Fig. 4A).

**Reviewer #2 (Public review):**
Summary:The authors observed gene ontologies associated with upregulated KLF2 target genes in HIV-1 RNA+ CD4 T Cells using scRNA-seq and scATAC-seq datasets from the PBMCs of early HIV-1-infected patients, showing immune responses contributing to HIV pathogenesis and novel targets for viral elimination.Strengths:The authors carried out detailed transcriptomics profiling with scRNA-seq and scATAC-seq datasets to conclude upregulated KLF2 target genes in HIV-1 RNA+ CD4 T Cells.

We thank the reviewer for highlighting the strengths of our work.

Weaknesses:This key observation of up-regulation KLF2 associated genes family might be important in the HIV field for early diagnosis and viral clearance. However, with the limited sample size and in-vivo study model, it will be hard to conclude. I highly recommend increasing the sample size of early HIV-1-infected patients.

Thank you to Reviewer #2 for this important comment. We acknowledge the limitations of our modest sample size, which reflects the challenges of recruiting well-characterized individuals in early HIV-1 infection (<6 months) and obtaining high-quality PBMCs for single-cell multi-omic profiling. To strengthen our findings, we validated the upregulation of KLF2 target genes using a publicly available scRNA-seq dataset from HIV-1-infected PBMCs [1], which showed similar expression patterns in HIV-1 RNA+ CD4+ T cells (page 8 and Supplementary Fig. 4A).

**Reviewer #3 (Public review):**
Summary:This manuscript studies intracellular changes and immune processes during early HIV-1 infection with an additional focus on the small CD4+ T cell subsets. The authors used single-cell omics to achieve high resolution of transcriptomic and epigenomic data on the infected cells which were verified by viral RNA expression. The results add to understanding of transcriptional regulation which may allow progression or HIV latency later in infected cells. The biosamples were derived from early HIV infection cases, providing particularly valuable data for the HIV research field.Strengths:The authors examined the heterogeneity of infected cells within CD4 T cell populations, identified a significant and unexpected difference between naive and effector CD4 T cells, and highlighted the differences in Th2 and Th17 cells. Multiple methods were used to show the role of the increased KLF2 factor in infected cells. This is a valuable finding of a new role for the major transcription factor in further disease progression and/or persistence.The methods employed by the authors are robust. Single-cell RNA-Seq from PBMC samples was followed by a comprehensive annotation of immune cell subsets, 16 in total. This manuscript presents to the scientific community a valuable multi-omics dataset of good quality, which could be further analyzed in the context of larger studies.

We sincerely thank the reviewer for the insightful and concise summary of our work.

Weaknesses:Methods and Supplementary materialsSome technical aspects could be described in more detail. For example, it is unclear how the authors filtered out cells that did not pass quality control, such as doublets and cells with low transcript/UMI content. Next, in cell annotation, what is the variability in cell types between donors? This information is important to include in the supplementary materials, especially with such a small sample size. Without this, it is difficult to determine, whether the differences between subsets on transcriptomic level, viral RNA expression level, and chromatin assessment are observed due to cell type variations or individual patient-specific variations. For the DEG analysis, did the authors exclude the most variable genes?

Thank you to Reviewer #3 for these detailed comments and observations. In the revised Methods section (page 16), we have added information on our quality control filtering process. Specifically, we excluded cells with fewer than 200 detected genes, high mitochondrial content (>30%), or low UMI counts. Doublets were identified and removed using DoubletFinder.

To address inter-donor variability, we included a new supplementary figure (Supplementary Fig. 1B) showing the distribution of major immune cell types across individual donors. While we observed some variation in cell-type composition between individuals, this likely reflects natural biological heterogeneity in early HIV-1 infection. Additionally, we applied fastMNN batch correction to mitigate donor-specific technical variation. After correction, the overall patterns of gene expression within each major CD4+ T cell subset were consistent across individuals (Supplementary Fig. 1C).

Regarding the DEG analysis, we used ‘FindMarkers’ function in Seurat (v.3.2.1), which does not exclude highly variable genes. These details have been clarified in the updated Methods section (page 18).

The annotation of 16 cell types from PBMC samples is impressive and of good quality, however, not all cell types get attention for further analysis. It’s natural to focus primarily on the CD4 T cells according to the research objectives. The authors also study potential interactions between CD4 and CD8 T cells by cell communication inference. It would be interesting to ask additional questions for other underexplored immune cell subsets, such as: (1) Could viral RNA be detected in monocytes or macrophages during early infection? (2) What are the inferred interactions between NK cells and infected CD4 T cells, are interactions similar to CD4-CD8 results? (3) What are the inferred interactions between monocytes or macrophages and infected CD4 T cells?

In line with our study objectives, we initially focused on CD4+ T cells as primary HIV-1 targets. However, in response to the reviewer’s comment, we examined the inferred communications between HIV-1-infected CD4+ T cells and other immune cells.

(1) With regard to the presence of viral RNA in monocytes or macrophages, we observed negligible HIV-1 RNA signal in these cell types in our dataset, consistent with their low permissiveness in early-stage infection [2]. However, we acknowledge the limitations of detecting rare infected cells at the single-cell level.

(2) We identified increased MIF and ICAM2 signaling between NK cells and HIV-1-infected CD4+ T cells, which are associated with KLF2-mediated immune modulation. These patterns are consistent with the CD4–CD8 interaction results observed in our dataset. (Supplementary Fig. 5A)

(3) Through the cell-cell interaction analysis with differential expression analysis, we inferred reduced CCL5 and CD55 signaling between monocytes and HIV-1-infected CD4+ T cells (Supplementary Fig. 5B). These reductions may potentially impair immune responses and antiviral defense.

We appreciate the reviewer’s suggestions and believe that the analysis of underexplored immune subsets strengthens the relevance of our findings. These results have been incorporated into the revised Results (page 9).

DiscussionIt would be interesting to see more discussion of the observation of how naïve T cells produce more viral RNA compared to effector T cells. It seems counterintuitive according to general levels of transcriptional and translational activity in subsets.Another discussion block could be added regarding the results and conclusion comparison with Ashokkumar et al. paper published earlier in 2024 (10.1093/gpbjnl/qzae003). This earlier publication used both a cell line-based HIV infection model and primary infected CD4 T cells and identified certain transcription factors correlated with viral RNA expression.

Thank you to Reviewer #3 for the insightful suggestions. We observed that the proportion of HIV-1-infected naïve CD4 T cells is higher compared to effector T cells. Although effector CD4 T cells are generally more active, previous studies have suggested that naïve CD4 T cells are susceptible to HIV-1 infection during early infection that may associate with initial expansion and rapid progression [3, 4]. This may be due to less restriction by antiviral signaling or more accessible chromatin states in resting cells. We have added this context and cited relevant papers to address this observation (page 11)

In addition, we have incorporated a comparative discussion with the recent study [5], which identified FOXP1 and GATA3 as transcriptional regulators associated with HIV-1 RNA expression. While these TFs were not significantly differentially expressed in our dataset, we discuss potential reasons for this discrepancy—including differences in infection model (in vitro vs. ex vivo), infection stage (latency vs. acute), and T cell subset composition—and emphasize that both studies highlight the importance of transcriptional regulation in HIV-1 persistence (page 12 and Supplementary Fig. 4B).

**Recommendations for the authors:**

**Reviewer #1 (Recommendations for the authors):**
The study has several notable limitations.First, it was restricted to early-stage HIV-1 infection (<6 months) without longitudinal data, preventing the authors from capturing temporal changes in immune cell populations, gene expression profiles, and epigenetic landscapes throughout disease progression.

Thank you to Reviewer #1 for this important limitation. As noted, our study focused exclusively on early-stage HIV-1 infection (<6 months) to capture the initial immune dysregulation and epigenetic alterations. We agree that longitudinal analysis would provide valuable insights into disease progression. However, due to the limited availability of early-infection patient samples suitable for performing multi-omics profiling, we prioritized capturing a detailed snapshot at this early stage. To address this limitation, future studies incorporating longitudinal sampling—including chronic infection and long-term non-progressors—will be essential to fully elucidate the temporal dynamics of HIV-1 pathogenesis.

Second, while the bioinformatic analysis compared "Uninfected" and "HIV-1-infected" cells from patients, the authors could have strengthened their findings by incorporating publicly available single-cell data from healthy donors and chronically infected HIV-1 patients to validate their arguments across all figures.

To support the robustness of our findings, we incorporated a publicly available single-cell RNA-seq dataset [1], which includes both healthy donors and individuals with chronic HIV-1 infection. In this dataset, we validated the upregulation of KLF2 and its target genes in HIV-1-infected CD4+ T cells and observed generally consistent expression patterns with those in our early-infection cohort (page 8; page 12 and Supplementary Fig. S4). While not all gene-level trends were identically reflecting differences in infection stage and immune activation status, this external comparison reinforces the reproducibility of key observations and highlights the unique transcriptional features associated with early HIV-1 infection.

Third, although the study focused on CD4+ T cells as primary HIV-1 targets, it overlooked other important immune cells such as CD8+ T cells, monocytes, and NK cells, which may contribute to viral persistence and immune dysfunction through cell-cell interactions.

In the revised manuscript, we expanded our analysis to include predicted ligand–receptor interactions between HIV-1-infected and uninfected CD4+ T cells with innate and cytotoxic immune cells using CellChat v.2.1.1. Specifically, we evaluated interactions with NK cells and monocytes and identified altered signaling pathways such as MIF, ICAM2, CCL5, and CLEC2B, which are associated with immune modulation (Supplementary Fig. 5A). We have added these results to the revised Results (page 9).

Lastly, comparing these findings with other chronic viral infections (e.g., HBV, HCV) would have positioned this work more effectively within the broader field of viral immunology and enhanced its impact.

We agree that broader comparisons with other chronic viral infections could enhance the impact of our findings. In the current discussion, we noted similarities in interferon signaling disruption with viruses such as HCV and HSV. (page 11). Our observation that HIV-1-infected CD4+ T cells exhibit impaired interferon responses is consistent with immune evasion mechanisms reported in HCV and HSV infections. These results underscore both the shared and specific features of immune modulation and persistence during HIV-1 early infection.

**Reviewer #3 (Recommendations for the authors):**
Supplementary Table S1 should indicate which technique was used for sequencing. However, the current version of the table marks no protocol applied to the majority of the samples, which is confusing and needs to be corrected.

Thank you to Reviewer #3 for pointing out this important oversight. We have revised Supplementary Table S1 to clearly indicate the sequencing method used for each sample. Separate columns for scRNA-seq, scATAC-seq, and sc-Multiome now specify whether each technique was applied (“Yes” or “No”) to improve clarity and transparency.

(1) Wang, S., et al., An atlas of immune cell exhaustion in HIV-infected individuals revealed by single-cell transcriptomics. Emerg Microbes Infect, 2020. 9(1): p. 2333-2347.

(2) Arfi, V., et al., Characterization of the early steps of infection of primary blood monocytes by human immunodeficiency virus type 1. J Virol, 2008. 82(13): p. 6557-65.

(3) Douek, D.C., et al., HIV preferentially infects HIV-specific CD4+ T cells. Nature, 2002. 417(6884): p. 95-8.

(4) Jiao, Y., et al., Higher HIV DNA in CD4+ naive T-cells during acute HIV-1 infection in rapid progressors. Viral Immunol, 2014. 27(6): p. 316-8.

(5) Ashokkumar, M., et al., Integrated Single-cell Multiomic Analysis of HIV Latency Reversal Reveals Novel Regulators of Viral Reactivation. Genomics Proteomics Bioinformatics, 2024. 22(1).